# A New Insight into an Alternative Therapeutic Approach to Restore Redox Homeostasis and Functional Mitochondria in Neurodegenerative Diseases

**DOI:** 10.3390/antiox11010007

**Published:** 2021-12-21

**Authors:** Dong-Hoon Hyun, Jaewang Lee

**Affiliations:** Department of Life Science, Ewha Womans University, Seoul 03760, Korea; Virtuoso0807@naver.com

**Keywords:** neurodegenerative diseases, oxidative stress, neuroinflammation, mitochondrial dysfunction, mitochondrial biogenesis, mitochondrial dynamics, plasma membrane redox enzymes

## Abstract

Neurodegenerative diseases are accompanied by oxidative stress and mitochondrial dysfunction, leading to a progressive loss of neuronal cells, formation of protein aggregates, and a decrease in cognitive or motor functions. Mitochondrial dysfunction occurs at the early stage of neurodegenerative diseases. Protein aggregates containing oxidatively damaged biomolecules and other misfolded proteins and neuroinflammation have been identified in animal models and patients with neurodegenerative diseases. A variety of neurodegenerative diseases commonly exhibits decreased activity of antioxidant enzymes, lower amounts of antioxidants, and altered cellular signalling. Although several molecules have been approved clinically, there is no known cure for neurodegenerative diseases, though some drugs are focused on improving mitochondrial function. Mitochondrial dysfunction is caused by oxidative damage and impaired cellular signalling, including that of peroxisome proliferator-activated receptor gamma coactivator 1α. Mitochondrial function can also be modulated by mitochondrial biogenesis and the mitochondrial fusion/fission cycle. Mitochondrial biogenesis is regulated mainly by sirtuin 1, NAD^+^, AMP-activated protein kinase, mammalian target of rapamycin, and peroxisome proliferator-activated receptor γ. Altered mitochondrial dynamics, such as increased fission proteins and decreased fusion products, are shown in neurodegenerative diseases. Due to the restrictions of a target-based approach, a phenotype-based approach has been performed to find novel proteins or pathways. Alternatively, plasma membrane redox enzymes improve mitochondrial function without the further production of reactive oxygen species. In addition, inducers of antioxidant response elements can be useful to induce a series of detoxifying enzymes. Thus, redox homeostasis and metabolic regulation can be important therapeutic targets for delaying the progression of neurodegenerative diseases.

## 1. Introduction

Neurodegenerative diseases comprise a wide range of diseases with heterogeneous aetiologies and exhibit degenerative processes commonly accompanied by oxidative stress and mitochondrial dysfunction [1]. Mitochondrial dysfunction is a major risk factor associated with aging and the initiation and progression of neurodegenerative diseases, such as Alzheimer’s disease (AD), Parkinson’s disease (PD), amyotrophic lateral sclerosis (ALS), and Huntington’s disease (HD). Neurodegenerative diseases are characterised by irreversible, progressive loss of neuronal cells, formation of protein aggregates, and a decline in cognitive or motor functions [2]. Neurodegenerative diseases are induced by imbalanced redox homeostasis and impaired energy metabolism [3], as hypothesised by several aging theories, including the free radical theory [4], the mitochondrial dysfunction theory [5], the genetic theory [6], and the telomere shortening theory [7].

The brain has a high demand for molecular oxygen and consumes about 20% of inhaled oxygen to maintain its function. More than 50% of the ATP produced in the brain is used to restore the resting membrane potential coupled to the Na^+^/K^+^ ATPase pump [8]. The brain contains large amounts of transition metals (e.g., copper and iron), which are responsible for the production of reactive oxygen species (ROS) [3]. In addition, brain cell membranes are enriched with polyunsaturated fatty acids, which are prone to lipid peroxidation. However, levels of antioxidant enzymes/molecules are relatively lower than in other organs. As a result, the brain is more sensitive to oxidative stress than any other part of the body.

The brain uses around 20% of the body’s glucose-derived energy and relies heavily on mitochondrial ATP production [9]. Therefore, normal brain mitochondrial function is required to maintain crucial physiological processes, such as synaptic transmission. Because deficits in mitochondrial function have been identified in many neurodegenerative diseases, maintenance of normal mitochondrial function during aging can be a way to prevent the progression of neurodegenerative diseases. In addition, the inflammatory process is identified to be closely associated with multiple pathways of neurodegenerative diseases. Inflammatory responses in the peripheral system can lead to consequent neuroinflammation and neurodegeneration [10]. This review discusses mitochondrial dysfunction associated with oxidative stress, neuroinflammation, and metabolic regulation, and will suggest a new approach to prevent the progression of neurodegenerative diseases.

## 2. Oxidative Stress and Redox Enzymes in Neurodegenerative Diseases

### 2.1. Alzheimer’s Disease

AD is the most common neurodegenerative disease affecting the elderly population and is characterised by selective, progressive death of cholinergic neurones, leading to the loss of cognitive functions and behavioural impairment. AD is an age-related disease, but can also be found in some young populations. The pathology of AD includes two types of protein aggregates, extracellular senile plaques containing amyloid β (Aβ) and intracellular neurofibrillary tangles formed from hyperphosphorylated tau [11,12]. Along with tau, the accumulation of oligomerised Aβ peptides mediates inflammation in neuronal cells, causing neurodegeneration. These protein aggregates induce deterioration in synaptic transmission, cholinergic denervation, and depleted acetylcholine.

Transition metals, such as iron, zinc, and copper, are known to produce ROS in cells. Aβ interacts with transition metals and is responsible for normal cellular signalling. However, Aβ can be aggregated through complexing with redox active copper [13]. Tau also is aggregated and phosphorylated after binding to zinc and iron [14]. High zinc levels in the neocortex and hippocampus in AD patients show the key role of zinc in redox homeostasis in the affected brain areas [15,16]. Recently, the putative role of iron in AD has been examined. Treatment with iron chelator improves cognitive capability, reducing Aβ aggregation and tau hyperphosphorylation in AD mouse model [17,18]. However, there remains a dilemma about the use of iron chelators because of iron’s significance in energy metabolism. Iron-sulphur clusters are essential factors for electron transfer in mitochondrial respiratory complex I, II, and III [19,20,21]. The Fenton reaction raises cellular ROS levels in the condition of high iron, whereas the low levels of iron decrease mitochondrial activity [22,23]. A recent study shows that lipid peroxidation promoted by the Fenton reaction leads to a new type of cell death, called ferroptosis [24]. AD post-mortem studies demonstrate typical features of ferroptotic cell death, including the increase of 4-HNE and the decrease of glutathione [25,26,27]. Inflammation in response to the formation of Aβ aggregates disrupts zinc homeostasis, leading to the release of zinc from the cerebrum and increased oxidative stress [28]. Although diverse functions of zinc make researching the mechanism between zinc and AD difficult, a recent study mentions that the supplement of zinc can reduce AD progression by lowering NLRP3-dependent inflammation [29]. As considered by an aforementioned study, low zinc may increase Aβ level in the brain of transgenic mice harbouring amyloid precursor protein with Swedish mutation and mutant human presenilin 1 (APP_Swe_/PS1ΔE9) [30].

ROS and oxidative stress play a crucial role in AD, as identified by oxidative stress induced by Aβ and oxidative damage, such as DNA/RNA oxidation (e.g., 8-hydroxydeoxyguanosine, 8-hydroxyganosine), protein oxidation (e.g., carbonylated proteins), and lipid peroxidation (e.g., 4-hydroxynoneal, malondialdehyde) [31,32,33]. Oxidative stress induced by accumulated Aβ inhibits complex IV activity, resulting in mitochondrial dysfunction and ATP depletion [31,32,33].

In addition, levels of antioxidants and antioxidant enzymes are decreased in AD models and patients with AD, suggesting an altered equilibrium between ROS production and antioxidant capacity. Vitamins C and E are decreased in the plasma of patients with mild cognitive impairment (MCI) or mild AD and in the cerebral spinal fluid of AD patients [34,35]. Glutathione (GSH) levels are also decreased in MCI and AD brains [36,37]. Glutathione S-transferase (GST), involved in GSH metabolism, is found in a modified carbonylated form in aged dog brains and *C. elegans* expressing Aβ [38,39] and in a nitrated form in MCI brains [40]. In particular, levels of superoxide dismutase (SOD), glutathione peroxidase, and catalase are decreased in the cortex of AD patients, whereas SOD levels (not activity) are increased in the hippocampus and amygdala [37]. Peroxiredoxins (Prxs), which remove hydrogen peroxide, are also affected by oxidative/nitrative stress. Prx2 oxidation is caused by Aβ in SAMP8 mice, while Prx2 level is increased in AD brains from SAMP8 mice and human [41,42]. Moreover, Prx6 is oxidatively modified in MCI brains [40].

At present, three choline esterase inhibitors (donepezil, rivastigmine, and galantamine) and one N-methyl-D-aspartate antagonist (memantine) approved by the Food and Drug Administration (FDA) have been used to treat AD in association with a Ginkgo biloba extract (EGb761, antioxidant) [43,44]. However, there is no known cure for AD. These drugs can delay AD progression, but induce common side-effects, including nausea, vomiting, and diarrhoea.

### 2.2. Parkinson’s Disease

PD is the second most common neurodegenerative diseases (ND) after AD in aged people. PD is characterised by the irreversible death of dopaminergic neurones in the substantia nigra (SN), causing postural instability, tremor, rigidity, and bradykinesia. The hallmark of PD is protein aggregates called Lewy bodies (LB) containing α-synuclein [45]. Mitochondrial dysfunction was explained first in PD pathogenesis as inhibition of mitochondrial complex I by 1-methyl-4-phenylpyridinium (MPP^+^), which is a metabolite of 1-methyl-4-phenyl-1,2,3,6-tetrahydropyridine (MPTP), causing parkinsonism [46]. Other toxic molecules, such as paraquat and 6-hydroxydopamine, can cause the symptoms of PD [47,48]. Oxidative stress and ROS are responsible for the pathogenesis of sporadic forms of PD. A high level of Fe^2+^ in the SN of PD patients promotes lipid peroxidation through the Fenton reaction, leading to nigral cell death [49]. Other transition metals (e.g., copper, zinc, and manganese) can cause neurodegeneration [50,51,52]. In addition, nitric oxide (NO) produced by neuronal nitric oxide synthase (nNOS) or inducible NOS (iNOS) inhibits mitochondrial complexes I and IV, resulting in enhanced production of ROS [53,54], consistent with enhanced levels of nNOS and iNOS in basal ganglia of post-mortem PD brains [55,56]. Deletions in mitochondrial DNA (mtDNA) have been found in the SN of elderly people and PD patients [57]. NO can induce lipid peroxidation by forming S-nitrosothiol compounds, resulting in PD phenotypes in mice treated with maneb and paraquat [58].

Familial cases of PD can be caused by various mutations in a number of genes, including α-synuclein, parkin, PTEN-induced kinase 1 (PINK1), DJ-1, and leucine-rich repeat kinase 2 (LRRK2) [59]. Both wild-type and mutant forms of α-synuclein aggregate during the progression of PD and are enriched in LB [60]. MPTP-treated transgenic mice exhibit overexpressed α-synuclein and dysfunctional mitochondria, resulting in nigral cell death [61]. Mutated parkin and PINK1 are related to the accumulation of dysfunctional mitochondria through reduced clearance of impaired mitochondria [62,63]. DJ-1 is a protein deglycase, prohibiting the aggregation of α-synuclein by functioning as a chaperone and an oxidative stress sensor [64,65]. DJ-1 protects neuronal cells against excessive oxidative stress. Mutations in DJ-1 are associated with autosomal recessive parkinsonism through multiple functions, such as an oxidative stress sensor and redox chaperone [66,67]. Mutated forms of LRRK2 increase cell sensitivity to mitochondrial inhibitors [68]. Therefore, the close relationships between oxidative stress, mitochondrial dysfunction, and accumulation of protein aggregates are key to PD pathogenesis.

At present, L-dopa (a natural precursor of dopamine) has been used with carbidopa, which blocks the conversion of L-dopa to dopamine outside the brain [69,70]. Safinamide, a monoamine oxidase B inhibitor, is used for patients with idiopathic PD [71]. In addition, antioxidants targeting the mitochondria can be effective for PD. Mitoquinone (MitoQ), which is ubiquinone conjugated to triphenylphosphonium (TPP), scavenges peroxyl, peroxynitrites, and superoxide radicals [72] and improves mitochondrial membrane potential (MMP) [73]. These drugs can improve PD symptoms but induce side effects such as fatigue and dizziness.

### 2.3. Amyotrophic Lateral Sclerosis

ALS, also called Lou Gehrig’s disease, is the most common type of motor neurone disease and is characterised by a progressive loss of motor neurones in the spinal cord, cortex, and brainstem. Oxidative stress, excitotoxicity, and inflammation are believed to be involved in ALS, although the links between them are not clear. A different type of protein aggregate (called Bunina bodies) also has been identified in ALS [74]. Mitochondrial dysfunction is an initiator of ALS. Mutations in Cu/Zn superoxide dismutase (SOD1) affect its antioxidant activity and cause accumulation of H_2_O_2_ and hydroxyl radicals, leading to the generation of impaired mtDNA and misfolded proteins [75]. Mutant SOD1 localises into the mitochondria and interacts with voltage-dependent anion-selective channel 1 (VDAC1), resulting in blockage of the exchange of ions and proteins between the mitochondria and cytosol [76]. Oxidative damage markers of DNA oxidation (e.g., 8-OHdG) and lipid peroxidation (e.g., isoprostane) have been identified in the brain of ALS patients [77]. ROS also cause mtDNA mutations, membrane permeability change, and impaired calcium homeostasis, leading to ALS [78,79].

Recently, two ALS drugs, riluzole (a glutamatergic neurotransmission inhibitor) and edaravone (an antioxidant drug), have been approved by the FDA [80].

## 3. Mitochondrial Dysfunction in Neurodegenerative Diseases

During oxidative phosphorylation, molecular oxygen can be converted to water through a series of electron transfers. Consequently, these processes can produce a variety of by-products, such as superoxide (O_2_^•−^), hydrogen peroxide (H_2_O_2_), and hydroxyl radical (OH^•^). Small amounts of ROS are generated in cells during normal energy metabolism. In physiological conditions, low-to-moderate levels of free radicals are involved in cellular homeostasis, signal transduction, synaptic plasticity, and immune response [81] (Figure 1). Excess amounts of free radicals can be harmful. Free radicals leaked from the electron transport chain (ETC) can attack DNA, lipids, and proteins in cells, leading to the generation of modified biomolecules through DNA oxidation, protein oxidation/nitration, and lipid peroxidation [82,83,84]. These altered molecules cause impairment of biochemical and physiological functions. Cells can detoxify these toxic molecules through the antioxidant defence system, such as SOD and reduced glutathione (GSH). When ROS generation overwhelms the antioxidant defence system (ADS), oxidative/nitrative stress induces pathophysiological conditions, including aging and age-related diseases [85].

The mitochondria are vulnerable to oxidative/nitrative stress. Mitochondrial DNA (mtDNA) is less packed than nuclear DNA, and the mtDNA repair system and antioxidant capacity are lower than those of cytosol [86]. In addition, the mitochondria are the primary site of ROS generation during energy metabolism. Therefore, mtDNA and mitochondrial proteins are more likely to be damaged by oxidative/nitrative stress, resulting in mitochondrial dysfunction, a common feature of many neurodegenerative diseases. Mutations in mtDNA (e.g., point mutations, large-scale deletions, tandem duplications) are identified in aged brains [87,88,89] and in patients with AD and PD [90,91].

In addition, the level of 8-hydroxyguanine in mtDNA is correlated with mtDNA deletions [92]. Altered mtDNA can cause not only transcription errors, but also the synthesis of proteins with impaired structures and functions. In addition, GSH and glutathione peroxidase levels are decreased in damaged mitochondria [93,94,95]. Abnormal mitochondrial complexes produced from mutated mtDNA and/or modified by oxidative/nitrative stress can induce defects in mitochondrial functions, such as attenuated complex I activity in AD, PD, and ALS [96,97]; defective complex II and IV activities in ALS [98]; and altered complex III activity in hearts of aged animals [99]. These impairments of mitochondrial activity can cause a shortage of ATP production [100,101], consistent with lower mitochondrial membrane potential in aged compared with young animals [102,103]. ATP depletion can affect subsequent biochemical processes.

Peroxisome proliferator-activated receptor gamma coactivator 1 α/β (PGC1α/β) is associated with mitochondrial dysfunction. PGC1α/β are involved in mitochondrial biogenesis, and their low levels are the key connectors between defective mitochondria and telomere shortening [104]. Increased p53 levels induced by DNA damage can inhibit PGC1α/β resulting in mitochondrial dysfunction and loss of SIRT1 activity, which is a protein functioning as NAD-dependent deacetylase [105,106,107,108,109].

Mitochondrial dysfunction found in a variety of neurodegenerative diseases can cause ATP shortage, resulting in impaired secondary biochemical cascades, such as cellular signalling and biosynthesis. As a result, progressive mitochondrial dysfunction can lead to progression of neural cell death, causing symptoms of AD, PD, and ALS.

## 4. Neuroinflammation in Neurodegenerative Diseases

The brain is a unique organ with an innate and acquired immunity tightly regulated in association with the peripheral system. The blood–brain barrier (BBB) can protect neurons against toxic chemicals and ordinary immune responses caused in the peripheral system. However, during the neuroinflammation process, the BBB becomes permeable and makes the brain sensitive to activated immune responses [110]. Following a viral infection or injection of lipopolysaccharide, microglia are first activated, causing the production of pro-inflammatory cytokines that promote permeabilization of the blood–brain barrier. Subsequently, leukocytes, T cells, and macrophages can be infiltrated into the brain. Impaired helper (CD4^+^) and cytotoxic (CD8^+^) T cells have been identified in the peripheral system of patients with neurovegetative diseases, suggesting that T cells can be involved in processes of neurodegeneration through a persistent antigenic challenge [10]. Acute neuroinflammation can be beneficial in response to brain injury by stimulating innate immunity [111]. However, chronic neuroinflammation is harmful to the brain because long-term activation of microglia releases inflammatory cytokines sustainably. These mediators can induce oxidative stress and cause the continuous inflammatory cycle [112], leading to prolonged inflammation, which is deleterious to many neurodegenerative diseases [113,114].

It has been shown that Aβ can directly cause neuronal cell death in AD brains and indirectly increase ROS production. Aβ aggregates can bind to microglia, causing the production of inflammatory cytokines, chemokines, and ROS [115]. Aβ aggregates, fibrillar Aβ, and neurofibrillary tangles (NFT) can activate the canonical pathway of complements (C3a, C3b, and C5a) [116]. Increased levels of pro-inflammatory cytokines (e.g., TNF-α, IL-6) were identified in the brain of AD patients compared with controls [117]. Many studies using mouse models have shown that Aβ deposition is promoted by activation of microglia and astrocytes [118]. Released inflammatory cytokines can up-regulate β-secretase [119], through the signalling pathway by TNF-α-activated nuclear factor kappa B (NF-κB), leading to increased Aβ production [120]. Neuroinflammation can enhance tau hyperphosphorylation (IL-6 via a cyclin-dependent kinase 5 pathway, IL-1 via MAPK pathway and NO) [121]. In addition, levels of prostaglandin D2 (PGD2; proinflammatory factor), PGE2, and PGJ2 were higher in the frontal cortex of AD patients than controls. PGD2 and its receptor were up-regulated in microglia and astrocytes of AD patients and mouse models [122]. Therefore, mechanisms involved in the aggregation of Aβ and tau, the consequent release of inflammatory cytokines, and increased ROS production can be therapeutic targets for slowing AD progression.

The administration of wild type and mutant α-synuclein initiated ROS production through the NADPH oxidase system [123], induced microglial activation and increased major histocompatibility complex (MHC) class II [124], and caused elevated IL1β, TNFα and IFNγ [125], and impaired immune profiles in central and peripheral systems [126]. The injection of a 6-hydroxy dopamine (6-OHDA) to rats causes ROS production, inducing up-regulation of pro-inflammatory cytokines (e.g., IL1β, IL6, TNFα, and IFNγ) and their receptors and down-regulation of anti-inflammatory mediators (e.g., IL10) [127]. Levels of DJ1 protein and its gene (PARK7) are higher in astrocytes from PD patients than in their control neurons [128]. Mutated DJ1 can alter lipid rafts responsible for the membrane receptor trafficking [129]. These findings show that chronic inflammation by environmental toxins and mutant proteins can aggravate inflammatory responses and then cause neurodegeneration.

The abnormal proliferation of astrocytes was identified in ALS patients and mouse models [130] and reactive astrocytes. Activated astrocytes expressed increased levels of COX2, inducible nitric oxide synthase (NOS), and neuronal NOS [130]. Aberrantly activated microglia secreted proinflammatory cytokines (e.g., TNFα, IL1β, and IL12), which can be neurotoxic to motor neurons [131]. Cytokines released by activated microglia (e.g., TNFα, IL1α) can induce the A1 subtype of reactive astrocytes, causing neuronal cell death in ALS [132].

Neuroinflammation is closely associated with oxidative stress in the pathogenesis of neurodegenerative diseases. Glial cells and infiltrated immune cells can produce a large amount of ROS in the brain [133]. Therefore, neuroinflammation is one of the causes of neurodegenerative disease through inducing oxidative stress and long-term activation of inflammatory processes.

## 5. Importance of Mitochondrial Biogenesis and Metabolic Regulation

The fact that dysfunctional mitochondria are identified in many neurodegenerative diseases suggests that maintaining mitochondrial function can be a good therapeutic target to delay the progress of such diseases. General approaches to treat diseases linked to altered mitochondria include conventional measures (e.g., optimised nutrition, dietary supplements) and symptom-based management. Growing evidence from mitochondria studies suggests therapeutic targets, such as mitochondrial biogenesis, associated with metabolic regulation and mitochondrial dynamics related to the fusion/fission cycle.

Mitochondrial biogenesis is a complex process driven by transcription factors and cofactors and the regulation of energy metabolism required for energetic demands in cells (Figure 2A). There are two ways to induce mitochondrial biogenesis (targeting upstream regulators and targeting downstream effectors), although it is difficult to separate the effects of mitochondria from those of other micro-organelles. Calorie restriction (CR) is the only reliable method to extend lifespan in a wide range of animals. CR induces crucial nutrient-sensing pathways, including those of sirtuins, NAD^+^, AMP-activated protein kinase (AMPK), mammalian target of rapamycin (mTOR), and peroxisome proliferator-activated receptor γ (PPARγ).

First, the sirtuin family (also called histone deacetylases) is a major target of CR and is activated by the elevated level of NAD^+^. Sirtuin 1 (SIRT1), a mammalian type of Sir2, stimulates the activity of various transcription factors and cofactors, such as the tumour suppressor p53 [134], myocyte-specific enhancer factor 2 (MEF2) [135], forkhead box transcription factors (FOXO) [136,137], and PPARγ coactivator 1-α (PGC1α) [108]. These transcription factors and cofactors are involved in mitochondrial biogenesis and functions. Resveratrol is not a direct inducer of SIRT1; instead, it stimulates AMPK to induce high NAD^+^ levels and activate SIRT1 indirectly [138,139]. Although several mechanisms are involved in rodent models and humans, resveratrol can increase mitochondrial biogenesis, lipid profiles, and insulin sensitivity [140,141].

Second, NAD^+^ is a cofactor for the sirtuin family and effectively balances mitochondrial function. Thus, the NAD^+^ level provides information about a cellular energy state. A high NAD^+^ level is modulated by increasing NAD^+^ synthesis [142] or transfer of electrons from NADH to other electron shuttles (e.g., oxidised coenzyme Q) [143]. In mice models, enhanced NAD^+^ level can be induced by supplementation with NAD^+^ precursors, such as nicotinamide mononucleotide (NMN) or nicotinamide riboside [144,145]. Interestingly, nicotinamide riboside stimulates the mitochondrial form of sirtuin, SIRT3 [144]. Similarly, inhibition of NAD^+^-consuming enzymes such as poly(ADP-ribose) polymerases (PARPs), or ADP ribosyl cyclase 1 (also called CD38) can elevate NAD^+^ level and SIRT1 activity, leading to the increased mitochondrial function and metabolic reprogramming [146,147]. In parallel, the mitochondrial function can be improved by overexpressing NADH-quinone oxidoreductase 1 (NQO1) or cytochrome by reductase (b5R), which transfer electrons from NADH to oxidised coenzyme Q in the plasma membrane [148,149]. Given that neurodegenerative disorders are associated with mitochondrial dysfunction by abnormal NAD^+^ levels [150,151,152,153], a method to increase cellular NAD^+^ levels would be great treatment strategies for neurodegenerative disorders.

Third, AMPK is a heterotrimeric complex comprising α, β, and γ subunits. Each of the subunits performs different functions (Figure 2B). AMPKα subunits have a conserved kinase domain that activates the whole AMPK. A recent study reported that the phosphorylation of Thr172 is essential for activating full AMPK [154]. Another recent work suggests that AMPKβ subunits maintain AMPK activation and lipid metabolism through interaction between carbohydrate binding module (CBM) and glycogen [155]. AMPKγ subunits play a significant role in energy-sensing of AMPK through cystathionine-β-synthase (CBS) domain [156]. AMPK activated by an increased AMP/ATP ratio can induce mitochondrial ATP production and mimic CR. AMPK as a metabolic regulator modulates metabolic processes and increases lipid peroxidation inhibiting malonyl CoA synthesis, suggesting that AMPK plays a crucial role in maintaining mitochondrial balance [157]. The AMPK agonist 5-aminoimidazole-4-carboxamide riboside activates a series of genes responsible for increasing exercise scores in wild-type mice [158] and attenuates mitochondrial dysfunction in cytochrome c-deficient mice [159]. Recent work also shows that two natural products (CMS121 and J147), Alzheimer’s disease (AD) drug candidates, maintain mitochondrial homeostasis and enhance neuroprotection in cells originated from aging mouse brains through inhibition of acetyl-CoA carboxylase (ACC)1 by AMPK [160].

Fourth, mTOR is a member of the phosphatidylinositol 3-kinase-related kinase family and acts as a sensor of levels of cellular nutrients, oxygen, and energy [161]. Inhibition of mTOR induces AMPK activation and enhanced mitochondrial functions. The treatment with rapamycin, an mTOR inhibitor, increases lifespan in many organisms [162,163]. Knockout of Raptor (a regulatory protein of mTOR) stimulates mitochondrial ATP production in adipose tissue [164].

Fifth, PPARγ is one of a small family of nuclear receptor genes involved in fatty acid oxidation and plays multiple roles in metabolic homeostasis. The heterodimerised part of PPARγ, retinoid X receptor α (RXR α), regulates mitochondrial retrograde signalling in cybrid cells with a mutation in tRNA_Leu_ [165]. This mutation causes decreased RXR α activity through JUN N-terminal kinase (JNK) activated by ROS. The deficient mitochondrial phosphorylation in these cybrid cells is attenuated by retinoic acid, an RXR agonist. Similarly, inhibition of corepressors of PPARγ, such as nuclear receptor corepressor 1 (Ncor1), increases the mitochondrial number and activity [135], consistent with decreased fat deposition in adipose tissue [166].

## 6. Alteration in Mitochondrial Dynamics and the Fusion/Fission Cycle in Neurodegenerative Diseases

The mitochondria are highly dynamic micro-organelles in cells and change their size, shape, and location through fission and fusion [167]. Fission and fusion are normal and continuous processes that occur in a variety of cells. Fission is mediated by the GTPase activity of dynamin-related protein 1 (Drp1) in the mitochondrial outer membrane. Drp1 promotes mitochondrial division through the creation of chains in association with mitochondrial dynamics protein 49 (MiD49) and mitochondrial dynamics protein (MiD51) complexed with mitochondrial fission factor (Mff) and mitochondrial fission protein 1 (Fis1) [168]. Fusion is facilitated by optic atrophy protein 1 (Opa1), mitofusin 1 (Mfn1), and mitofusin 2 (Mfn2) [168].

The fusion/fission cycle in mitochondria is a balanced system involving the concerted and sequential binding of fusion proteins (e.g., Mfn1, Mfn2, Opa1) and fission proteins (e.g., Drp1, Fis1) (Figure 3). Interestingly, a recent study suggests mitochondrial DRP1 receptors (e.g., MFF, MID49, MID51, and FIS1) determine mitochondrial destiny [169]. Especially, MFF promoted midzone scission events that led to mitochondrial proliferation by interacting with the endoplasmic reticulum (ER). By contrast, FIS1 participates in degrading a small part of mitochondria at the peripheral region, contacting the lysosome [170]. However, the role of FIS1 in mitochondrial life has been a controversial issue since its depletion has a marginal effect on mitochondrial division [169]. In addition, other perspectives mainly focus on AMPK, which could directly control mitochondrial homeostasis [171]. Especially, one group demonstrates that inositol can inhibit mitochondrial fission under energy stress by directly targeting AMPKγ subunits [172]. These recent studies show that diverse factors participate in maintaining mitochondrial homeostasis.

The proportion of these fusion or fission proteins involved in mitochondrial quality control is regulated in response to ROS levels. When mitochondrial antioxidant enzymes (e.g., SOD, Gpx, Prx) are not sufficient and ROS production is excessive, ROS produced in the mitochondria can damage mitochondrial biomolecules and cause the loss of membrane potential and impair ATP generation, resulting in further ROS production. DNA repair enzymes and lipases can restore oxidatively damaged biomolecules. Alternatively, mitochondrial efficiency can be restored through the fusion cycle [173]. It is hypothesised that mitochondria can regulate their contents during the fusion cycle to dilute them and facilitate their repair system. However, when these protective mechanisms are not fully functional, damaged biomolecules in the mitochondria can affect the other parts of the micro-organelle, causing the removal of damaged mitochondria via mitophagy in the fission process. Accumulation of damaged mitochondria induces mitochondrial fragmentation, leading to apoptosis.

Mitochondrial dynamics are altered in many neurodegenerative diseases (Figure 3). The dysregulation of proteins involved in the mitochondrial fusion/fission cycle is associated with several neurodegenerative diseases. Increased levels of Drp1 and Fis1 and decreased levels of Opa1, Mfn1, and Mfn2 have been identified in the frontal cortex of AD patients [174]. In addition, increased mitochondrial fusion was found in AD mice with knock-in Drp1+/− [175]. Excessive Drp1 causes increased mitochondrial fission in cells overexpressing mutant SOD-1 [176]. Changes in levels of Drp1/Fis1 and Opa1/Mfn1 induce loss of motor neurones in ALS models with mutant SOD-1 [177].

Mitophagy is a selective pathway controlling the number and quality of mitochondria under nutrient-enriched conditions, whereas autophagy is a non-selective recycling process of proteins and micro-organelles (e.g., endoplasmic reticulum, mitochondria) in response to nutrient-deficient stress [178]. Neuronal cells require essential mediator proteins responsible for mitophagy, such as parkin and PINK1. Impairment of mitochondrial membrane potential causes PINK1 accumulation at the surface of the mitochondria and parkin recruitment, which ubiquitinate mitochondrial outer membrane proteins for recognition by autophagosomes [179]. However, mutated parkin or PINK1 in PD cannot recruit parkin or parkin-mediated mitophagy, causing accumulation of dysfunctional mitochondria [180,181]. Overexpressed parkin or PINK1 can increase the fission process through ubiquitination of Mfn1 and Mfn2 [182,183].

Conventional approaches for improving mitochondrial function are using synthetic and semi-synthetic compounds. There are two approaches to treat diseases related to dysfunctional mitochondria: a target-based approach and a phenotype-based approach [184]. A target-based approach is relatively easy to apply because of its suitability for high-throughput screening platforms and conservation with human proteins. However, this approach has a considerable restriction in that a selected compound can bind to a molecule included in the multiple regulatory pathways linked to mitochondrial homeostasis. For example, several agonists that increase mitochondrial functions are associated with specific signalling molecules, such as SIRT1 (e.g., STAC-1, STAC-2) [185], AMPK (e.g., A769662) [186], and PPAR (e.g., FMOC-L-Leucine) [187]. These isolated targets have disadvantages in application to multi-organ physiology and genome-wide screening involving multiple pathways and relatively broad targets.

In contrast, a phenotype-based approach does not focus on a single pathway but modulates integrated pathways governing mitochondrial functions. A phenotype-based approach is challenging to design and apply, but it can help to identify novel proteins or pathways regulating mitochondrial functions or biogenesis. Immortalised cell lines exhibit metabolic reprogramming in response to energetic stress through stimulation of mTOR signalling and inhibition of the AMPK pathway [188]. As a result, immortalised cell lines lose the normal physiological characteristics of eukaryotic cells and adapt to new environments to survive. Despite this problem, cell lines are convenient to culture. They have been used to develop mitochondrial modulators, such as fusion promotors (e.g., M1 hydrazone) [189], metabolic regulators (e.g., meclizine) [190], and anti-mitotic drugs (e.g., podophyllotoxins) [184]. Primary cells are more similar to the in vivo system than are immortalised cells. Primary cells isolated from patients or disease models can be applied to treat genetic diseases. BRD6897 modulates mitochondrial biogenesis and turnover and might require the co-culture of different cell types to build complex systems [191].

## 7. The New Compensatory Mechanism in Response to Mitochondrial Dysfunction

Cells have compensatory mechanisms for cell survival in response to mitochondrial dysfunction. Limited ATP supply can be caused by strenuous muscle activity (e.g., extreme exercise) or mitochondrial dysfunction, which is identified in many neurodegenerative diseases and cells lacking functional mitochondria. Under conditions of energy restriction, cells can generate additional ATP by enhanced glycolysis coupled to lactate fermentation. Mitochondria-deficient cells (also called ρ^o^ cells) can survive by producing more ATP through increased glycolysis linked to activated electron transport in the plasma membrane when treated with pyruvate and uridine [192,193]. In addition, ρ^o^ cells show decreased production of ROS and increased activity of plasma membrane redox enzymes, such as NQO1 and b5R compared with the parental cells [194].

Plasma membrane (PM) redox enzymes ubiquitous in all types of eukaryotic cells are involved in normal cellular physiology and redox homeostasis [143,195]. Previous studies showed protective roles, such as quinone detoxification, O_2_^•−^ scavenging, p53 stabilisation, and reduction of oxidised α-tocopherol of PM redox enzymes in neuronal cells in response to oxidative and energetic stress (Figure 4). PM redox enzyme activity is elevated in lymphocytes from patients with insulin-dependent diabetes mellitus, whose representative hallmark is diminished mitochondrial function [196]. When supplemented with a reduced form of coenzyme Q (CoQ), PM redox enzymes delayed apoptotic cell death in response to oxidative stress in AD brains through maintenance of redox homeostasis [197,198]. Interestingly, PM redox enzymes can be involved in extending lifespan in yeast and mammals by enhancing the NAD^+^/NADH ratio and activating SIRT1 [199,200].

In contrast, impaired PM redox enzymes and other related compounds are identified in neurodegenerative diseases. In the hippocampal neurons of triple transgenic mice containing presenilin 1 (M146V), APP_Swe_ and tau (P301L) transgenes, which lead to amyloid β plaques and neurofibrillary tangles [201], NQO1 expression is lower than in age-matched controls [202]. A missense mutation decreases a heterozygous NQO1 activity in codon 187 due to a C609T polymorphism in the NQO1 cDNA [203,204]. High levels of C/T and T/T alleles found in AD patients suggest a low level of the C/C allele as a risk factor for AD [205]. Levels of PM components (e.g., cholesterol, sphingomyelin) are altered in AD [206]. Levels of oxidised α-tocopherol are significantly enhanced, and total α-tocopherol content is diminished in AD and vascular dementia [207,208]. In addition, altered PM enzyme activity and decreased levels of CoQ and α-tocopherol are identified in the triple transgenic mice [209]. These findings suggest that up-regulated PM redox enzymes can delay symptoms of neurodegenerative diseases, including mitochondrial dysfunction. In fact, PM redox enzymes are activated by CR [210,211]. Increased activity of mitochondrial complexes, higher ATP production, and lower ROS generation were induced by overexpressed NQO1 or b5R, suggesting efficient electron transport in the mitochondrial complexes [149]. Similarly, the enhanced mitochondrial complex functions, the diminished oxidative damage, and the modest lifespan extension were identified in mice overexpressing b5R [212].

In particular, NQO1 can be a good therapeutic target for neurodegenerative diseases, including AD, because it is an inducible enzyme responsible for 2-electron transfer without the production of semi-quinone radicals [213,214]. This protein is expressed by the antioxidant response element (ARE) associated with nuclear factor erythroid-2-related factor 2 (Nrf2) and nuclear factor kappa-light-chain-enhancer of activated B cells (NF-κ_B_) [215] (Figure 4). Under normal conditions, disulphide bonds between Nrf2 and Keap1 remain stable, and the Nrf2-Keap1 complex moves to the 26S proteasome, where it is degraded. However, oxidative stress breaks the disulphide bonds, allowing Nrf2 to be phosphorylated and translocated into the nucleus. The translocated Nrf2 binds to the ARE and transcriptional coactivators CREB binding protein (CBP)/p300 in association with FOXO3, expressing a series of detoxifying enzymes, including NQO1, Prx1, and heme oxygenase 1 (HO-1) [216]. A cytosolic form of NQO1 can translocate into the inner surface of the PM in response to oxidative/metabolic stress.

In addition, natural compounds, such as phytochemicals, which activate ARE expression, can be good therapeutic inducers because of their various protective effects against toxic insults. Sulforaphane increases the resistance of retinal cells to UV-induced photooxidative damage [217]. Dietary supplementation with curcumin induces lower ischaemic damage in gerbils and decreases the level of Aβ in transgenic mice with APP_Swe_ through reduced oxidative damage and inflammation by HO-1 and p38 MAP kinase [218,219]. Curcumin protects neuronal cells against toxic insults and blocks the formation of Aβ plaques [220]. Allicin and allium protect hippocampal neurones from Aβ and tunicamycin via increasing levels of uncoupling proteins [221], decreasing oxidative stress by stimulating the Nrf2-ARE pathway [222]. Some cytotoxic effects of lipid-soluble ginseng extracts can be attenuated by the overexpression of NQO1 [223]. These findings suggest that the induction of detoxifying enzymes, including NQO1 through the Nrf2-Keap1 pathway, can be a good approach to improve mitochondrial function and delay the progression of neurodegenerative diseases.

## 8. Conclusions

The brain acquires nutrients and molecular oxygen to produce ATP in the mitochondria. However, the brain has a relatively low antioxidant capacity and consumes high amounts of oxygen in the mitochondria. In the early stages of neurodegenerative diseases, oxidative/metabolic stress and neuroinflammation induce mitochondrial dysfunction and dysregulated immune responses in neuronal cells, consistent with altered mitochondrial dynamics. Consequently, mitochondrial dysfunction accelerates the progression of neurodegenerative diseases. Neuronal cells can survive following activation of mitochondrial biogenesis or other compensatory mechanisms, such as PM redox enzymes. A high NAD^+^/NADH ratio is a primary factor responsible for maintaining mitochondrial functions and biogenesis through SIRT1, AMPK, mTOR, and PPARs.

Therefore, one of the most effective therapeutic approaches for neurodegenerative diseases is maintaining mitochondrial functions and biogenesis because it is the best way to delay the progression of neurodegenerative diseases at an early stage. Elevating NAD^+^ level and ATP production without further ROS production, through NQO1 activation can be a good target for many neurodegenerative diseases. Taken together, these results indicate that specific ARE induction, stimulated cell survival signalling, and improved mitochondrial function can be a promising therapeutic strategy for the prevention and treatment of neurodegenerative diseases.

## Figures and Tables

**Figure 1 antioxidants-11-00007-f001:**
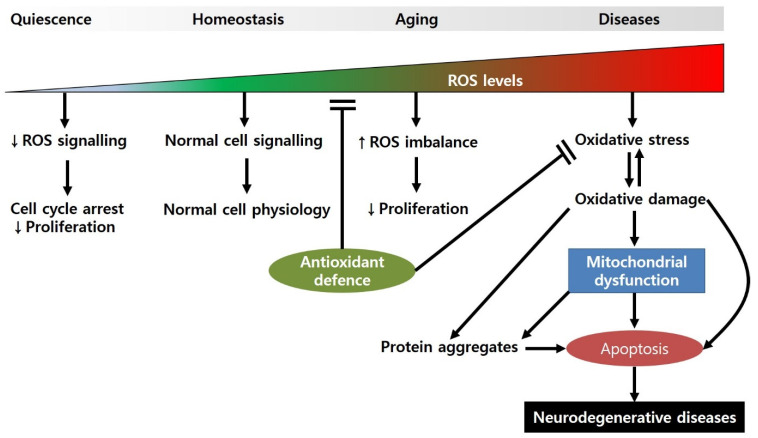
Regulation of cellular physiology by ROS levels. When ROS levels are very low, the cell cycle can be arrested, and proliferation is slowed. Under normal ROS levels, cells show normal cell physiology (e.g., cellular homeostasis, cell division, synaptic plasticity, etc.) by maintaining appropriate signalling pathways. However, during the aging processes, ROS production is increased due to the attenuated antioxidant defence, resulting in decreased proliferation. When ROS levels are high, they can induce oxidative-stress-induced damage to biomolecules, causing mitochondrial dysfunction and apoptotic cell death.

**Figure 2 antioxidants-11-00007-f002:**
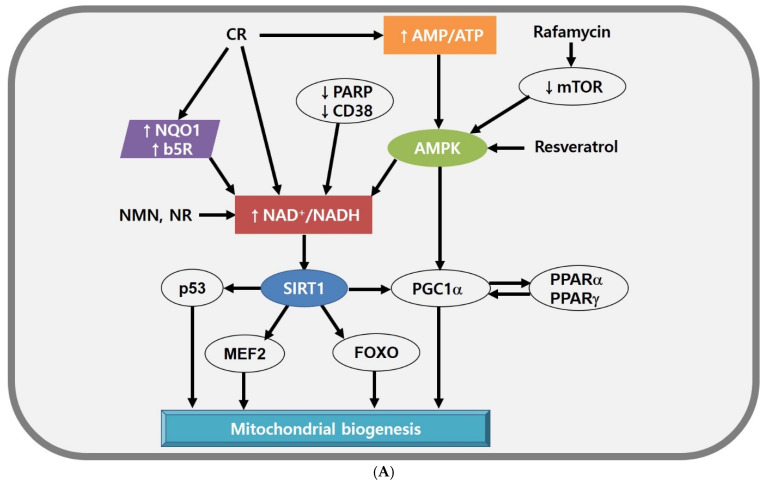
(**A**) Mitochondrial biogenesis modulated by SIRT1, NAD^+^, and AMPK. Energetic stress (e.g., high NAD^+^/NADH ratio, increased AMP level) is a good enhancer of mitochondrial biogenesis by activating SIRT1 and PGC1α. CR elevates NAD^+^ level directly or by activating PM redox enzymes or AMPK indirectly. Rafamycin and resveratrol can enhance AMPK activity. (**B**) The structure of AMPK and the function of AMPK subunits. Each of the subunits plays a different role in AMPK activation. AMPK α subunit activates the AMPK complex, AMPK β subunit stabilises the AMPK complex, and AMPK γ subunit serves as energy-sensor. The activated AMPK complex leads to lipid metabolic change and maintains mitochondrial homeostasis. Abbreviations: ACC, acetyl-CoA carboxylase; AMPK, AMP-activated protein kinase; b5R, cytochrome b5 reductase; CD38, ADP ribosyl cyclase 1; CR, calorie restriction; FOXO, forkhead box transcription factors; MEF2, myocyte-specific enhancer factor 2; mTOR, mammalian target of rapamycin; NQO1, NADH-quinone oxidoreductase 1; NR, nicotinamide riboside; PARP, poly(ADP-ribose) polymerases; PGC1α, PPARγ coactivator 1-α; PPAR, peroxisome proliferator-activated receptor; SIRT, Sirtuin 1.

**Figure 3 antioxidants-11-00007-f003:**
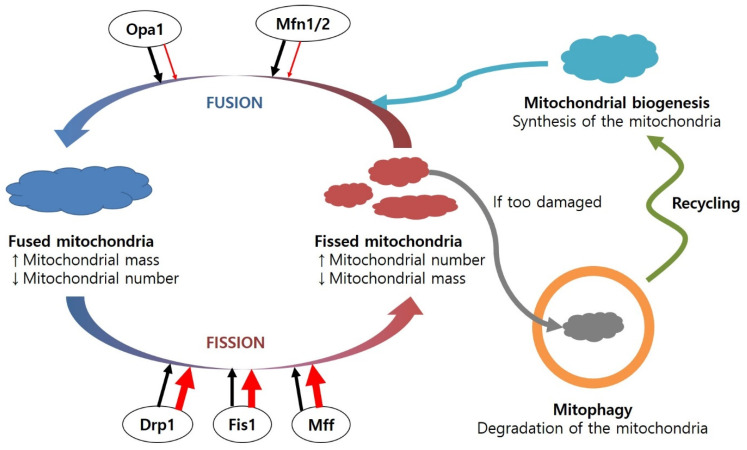
Regulation of the fusion and fission cycle in the mitochondria, including mitochondrial biogenesis and mitophagy. The mitochondrial fusion/fission cycle is balanced under healthy conditions. However, under diseased conditions, the levels of fusion proteins are decreased, while amounts of fission proteins are increased, resulting in a number of dysfunctional mitochondria. Damaged mitochondria can be combined with lysosomes and degraded. Abbreviations: Drp1, dynamin-related protein 1; Fis1, mitochondrial fission protein 1; Mff, mitochondrial fission factor; Mfn1/2, mitofucin 1/2; Opa1, optic atrophy protein 1.

**Figure 4 antioxidants-11-00007-f004:**
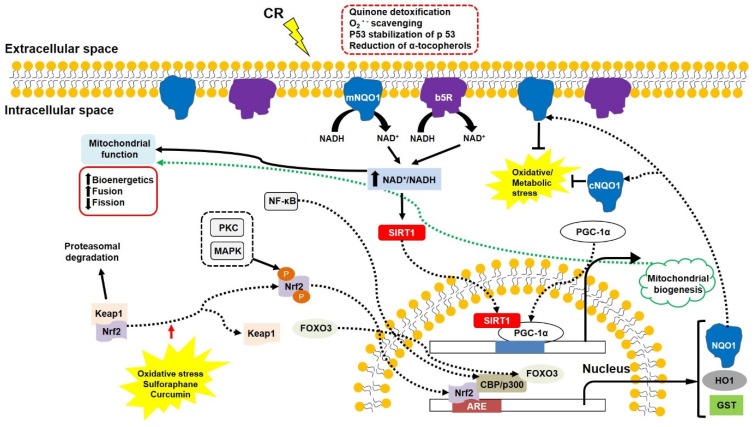
The PM redox enzymes (NQO1 and b5R) play a crucial role in increasing NAD^+^/NADH ratio and decreasing oxidative/metabolic stress. NAD^+^/NADH ratio is the key factor in inducing cell survival signalling involving SIRT1, PGC1α, and Nrf2. SIRT1 and PGC1α promote mitochondrial biogenesis, and Nrf2 induces ARE expression associated with p300 and FOXO3. Some phytochemicals can break the Nrf2–Keap1 linkage, inducing detoxifying enzymes. Abbreviations: ARE, antioxidant response element; b5R, cytochrome b5 reductase; CBP, transcriptional coactivators of CREB binding protein; CR, calorie restriction; FOXO3, O subclass 3 of the forkhead family of transcription factors. GST, glutathione S-transferase; HO1, heme oxygenase 1; Keap1, Kelch-like ECH-associated protein 1; MAPK, mitogen-activated protein kinase; NF-kb, nuclear factor kappa-light-chain-enhancer of activated B cells; NQO1, NADH-quinone oxidoreductase 1; cNQO1, cytosolic NQO1; mNQO1, membrane-bound NQO1; Nrf2, nuclear factor erythroid-2-related factor 2; PGC1α, peroxisome proliferator-activated receptor gamma coactivator 1-α; PKC, protein kinase C; SIRT1, silent mating type information regulation 2 homolog 1.

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
