# Peer review of "A New Insight into an Alternative Therapeutic Approach to Restore Redox Homeostasis and Functional Mitochondria in Neurodegenerative Diseases"

_antioxidants, 2021, doi:10.3390/antiox11010007_

Round 1
Reviewer 1 Report
Review of “The new insight into the alternative therapeutic approach to restore redox homeostasis and functional mitochondria in neurodegenerative diseases”
The authors present a comprehensive review on the effect of redox homeostasis on brain heath and neurodegenerative disease as well as the current state of treatments for such diseases. Three neurodegenerative diseases are discussed, Alzheimer’s disease, Parkinson’s disease and Amyotrophic Lateral Sclerosis and their relationship to oxidative stress. The association of these diseases with neuroinflammation and mitochondrial dysfunction are described in great detail. The reviewer found section 6 particularly interesting and informative.
The manuscript is well developed, organized, and referenced. The reviewer found mostly minor modifications that should be made to enhance the impact and make it more readable for those not as familiar with the field.
Line 81: “of” should be “with”
Line 86: “but rather” should be “whereas”
Line 93-94: Why are Ab levels increased? Is this due to high levels of metals? The context is unclear.
Line 111-112: Are all of these observations for SAMP8 mice? If not, the organism involved should be clearly stated.
Line131-132: How do the other metals cause neurodegeneration?
Line 146: Define DJ-1 and give it context.
Line 221: SIRT1 is not defined until line 315, the fourth time the abbreviation is used. SIRT1 should be defined here.
Line 231-234: What organisms were used in these studies?
Line 254: The fact that these prostaglandins are proinflammatory should be noted.
Line 287-288: The reviewer is unable to determine the intent of this sentence. Do mitochondria do this on their own? Is this denoting a new hypothesis?
Line 477-481: Are these referring to preclinical studies on rodents or human clinical studies?
Author Response
Following are our responses to comments of the Reviewer 1.
Reviewer #1
Review of “The new insight into the alternative therapeutic approach to restore redox homeostasis and functional mitochondria in neurodegenerative diseases”
The authors present a comprehensive review on the effect of redox homeostasis on brain heath and neurodegenerative disease as well as the current state of treatments for such diseases. Three neurodegenerative diseases are discussed, Alzheimer’s disease, Parkinson’s disease and Amyotrophic Lateral Sclerosis and their relationship to oxidative stress. The association of these diseases with neuroinflammation and mitochondrial dysfunction are described in great detail. The reviewer found section 6 particularly interesting and informative.
The manuscript is well developed, organized, and referenced. The reviewer found mostly minor modifications that should be made to enhance the impact and make it more readable for those not as familiar with the field.
- Line 81: “of” should be “with”
Response: As recommended, “of” has been changed to “with”.
- Line 86: “but rather” should be “whereas”
Response: As recommended, “but rather” has been changed to “whereas”.
- Line 93-94: Why are Ab levels increased? Is this due to high levels of metals? The context is unclear.
Response: There was no clear clue that zinc directly controls Ab levels. However, a recent study shows that the supplement of zinc can decrease AD progression (doi.org/10.1523/JNEUROSCI.1980-20.2020). As considered by the previous study, it can be assumed that the small amount of zinc may be related to high Ab levels in transgenic mice (APPSwe/PS1DE9). Thus, "lines 93-94" has been revised and the abobe new relavant reference are also added in the Reference.
- Line 111-112: Are all of these observations for SAMP8 mice? If not, the organism involved should be clearly stated.
Response: They used the mice model and human postmortem samples, respectively. The exact subjects used in the manuscript are clerly mentioned in this manuscript (page 3).
- Line131-132: How do the other metals cause neurodegeneration?
Response: These metals tend to directly link to specific proteins leading to aggregation and interaction with nucleic acids. This results in cell damage and death by inducing the formation of abnormal and toxic aggregates and incessantly consuming cellular antioxidants via Fenton and Haber–Weiss reactions that promote the redox cycle. The relavant references (doi.org/10.1007/s12017-009-8102-1; doi.org/10.1038/sj.bjp.0706416) have been added in the Reference section.
- Line 146: Define DJ-1 and give it context.
Response: The definition of DJ-1 was attached to the manuscript according to your suggestion (page 4) and added references into the Reference section (doi.org/10.1371/journal.pbio.0020362; doi.org/10.1016/j.jmb.2005.12.030).
- Line 221: SIRT1 is not defined until line 315, the fourth time the abbreviation is used. SIRT1 should be defined here.
Response: The definition of SIRT1 was added to the manuscript according to the reviewer‘s suggestion (page 5) and a relevant references is included in the Reference section (doi.org/10.1006/bbrc.1999.0897; doi.org/10.1038/sj.emboj.7600244; doi.org/10.1038/nature03354).
- Line 231-234: What organisms were used in these studies?
Response: Mice were used.
- Line 254: The fact that these prostaglandins are proinflammatory should be noted.
Response: As recommended, the sentence has been revised in the manuscript (page 6).
- Line 287-288: The reviewer is unable to determine the intent of this sentence. Do mitochondria do this on their own? Is this denoting a new hypothesis?
Response: We agree with the reviewr’s opinion, and the sentence seems to decrease the coherence of the contents. Thus, the sentence has been deleted.
- Line 477-481: Are these referring to preclinical studies on rodents or human clinical studies?
Response: Reference 197 implemented patient samples with insulin-dependent DM. References 198 and 193 only used cell lines. These show in vitro studies, not preclinical studies using animals or humans.
Reviewer 2 Report
The review by Dong-Hoon Hyun and Jaewang Lee focused on an alternative therapeutic approach to restore redox homeostasis and functional mitochondria in neurodegenerative diseases. The review is well written and very interesting. The explanatory graphics are well done and easy to understand. I do not require any changes. The work can be accepted in the form presented by the authors.
Author Response
Following are our responses to comments of the Reviewer 1.
Reviewer #2
The review by Dong-Hoon Hyun and Jaewang Lee focused on an alternative therapeutic approach to restore redox homeostasis and functional mitochondria in neurodegenerative diseases. The review is well written and very interesting. The explanatory graphics are well done and easy to understand. I do not require any changes. The work can be accepted in the form presented by the authors.
Response: Thanks for the reviewer’s comments.